# Heterologous Expression, Purification, and Immunomodulatory Effects of Recombinant Lipoprotein GUDIV-103 Isolated from *Ureaplasma diversum*

**DOI:** 10.3390/microorganisms10051032

**Published:** 2022-05-16

**Authors:** Manoel Neres Santos-Junior, Wanderson Souza Neves, Ronaldo Silva Santos, Palloma Porto Almeida, Janaina Marinho Fernandes, Bruna Carolina de Brito Guimarães, Maysa Santos Barbosa, Lucas Santana Coelho da Silva, Camila Pacheco Gomes, Beatriz Almeida Sampaio, Izadora de Souza Rezende, Thiago Macedo Lopes Correia, Nayara Silva de Macedo Neres, Guilherme Barreto Campos, Bruno Lopes Bastos, Jorge Timenetsky, Lucas Miranda Marques

**Affiliations:** 1Department of Biointeraction, Multidisciplinary Institute of Health, Federal University of Bahia, Vitória da Conquista 40170-110, Brazil; neres.manoel@hotmail.com (M.N.S.-J.); wandersonneves1234@gmail.com (W.S.N.); ronaldossantos0801@gmail.com (R.S.S.); jannaina.marinhom@gmail.com (J.M.F.); thiagomlc_94@yahoo.com.br (T.M.L.C.); naysmacedo@yahoo.com.br (N.S.d.M.N.); 2Department of Biology, and Biotechnology of Microorganisms, State University of Santa Cruz (UESC), Ilhéus 45662-900, Brazil; brunacarolina.biotec@gmail.com (B.C.d.B.G.); lucassantana864@gmail.com (L.S.C.d.S.); mylla.gomes@yahoo.com.br (C.P.G.); beatriz7s@hotmail.com (B.A.S.); guilhermebcampos@hotmail.com (G.B.C.); bastosbl@gmail.com (B.L.B.); 3Bioinformatics and Computational Biology Lab, Division of Experimental and Translational Research, Brazilian National Cancer Institute (INCA), Rio de Janeiro 20231-050, Brazil; pahporto@gmail.com; 4Department of Microbiology, Institute of Biomedical Science, University of São Paulo, São Paulo 05508-000, Brazil; maysabarbosa_06@hotmail.com (M.S.B.); izarezende90@hotmail.com (I.d.S.R.); joti@usp.br (J.T.)

**Keywords:** *Ureaplasma diversum*, lipoproteins, lymphoproliferation, immunomodulation

## Abstract

*Ureaplasma diversum* is a bacterial pathogen that infects cattle and can cause severe inflammation of the genital and reproductive systems. Lipid-associated membrane proteins (LAMPs), including GUDIV-103, are the main virulence factors in this bacterium. In this study, we heterologously expressed recombinant GUDIV-103 (rGUDIV-103) in *Escherichia coli*, purified it, and evaluated its immunological reactivity and immunomodulatory effects in bovine peripheral blood mononuclear cells (PBMCs). Samples from rabbits inoculated with purified rGUDIV-103 were analysed using indirect enzyme-linked immunosorbent assay and dot blotting to confirm polyclonal antibody production and assess kinetics, respectively. The expression of this lipoprotein in field isolates was confirmed via Western blotting with anti-rGUDIV-103 serum and hydrophobic or hydrophilic proteins from 42 *U. diversum* strains. Moreover, the antibodies produced against the *U. diversum* ATCC 49783 strain recognised rGUDIV-103. The mitogenic potential of rGUDIV-103 was evaluated using a lymphoproliferation assay in 5(6)-carboxyfluorescein diacetate succinimidyl ester–labelled bovine PBMCs, where it induced lymphocyte proliferation. Quantitative polymerase chain reaction analysis revealed that the expression of interleukin-1β, toll-like receptor (TLR)-α, *TLR2**, TLR4,* inducible nitric oxide synthase, and caspase-3–encoding genes increased more in rGUDIV-103–treated PBMCs than in untreated cells (*p* < 0.05). Treating PBMCs with rGUDIV-103 increased nitric oxide and hydrogen peroxide levels. The antigenic and immunogenic properties of rGUDIV-103 suggested its suitability for immunobiological application.

## 1. Introduction

*Ureaplasma diversum*, which is approximately 400–500 nm in diameter, is a facultative intracellular pleomorphic bacterium (coccoid or coccobacillary) belonging to the Mollicutes class and is associated with respiratory and reproductive tract pathologies in cattle. The cause of each disease is multifactorial, as with infections attributed to other members of Mollicutes [1,2,3,4]. The bacteria have a carbohydrate-like capsule outside the cytoplasmic membrane instead of a cell wall [5,6]. They are microaerophilic (8–10% of CO_2_) with 37 °C optimal growth temperature and 6.0–7.0 optimal pH. *U. diversum* produces relatively small colonies (100–175 μm in diameter) in solid medium [6,7,8]. *U. diversum* growth increases the pH of the liquid medium owing to the extracellular release of ammonia. This increase was revealed by an indicator present in the culture medium [9]. *Ureaplasma* (species) spp. require urea to produce adenosine triphosphate (ATP), which releases ammonia that may damage host tissues [10].

*U. diversum* is detected using culture techniques or polymerase chain reaction (PCR) [11,12,13,14,15]. It primarily infects the respiratory and genital/reproductive tracts of cattle [11,16,17]. Although granular vulvitis is common in infected cows, the infections are either symptomatic or asymptomatic [18,19,20,21]. Chronic granular vulvovaginitis may progress to endometritis and result in miscarriage or infertility if not diagnosed and treated [22,23,24]. *U. diversum* isolated from foetal lungs is associated with abortion and neonatal death in calves, and endobronchial inoculation assays indicate that it causes pneumonia in calves [25,26]. In bulls, *U. diversum* is associated with seminal vesiculitis, balanoposthitis, epididymitis, and other pathologies caused by morphological and functional changes in the sperm [1,23,27]. Semen infection and viable permanence in blastocysts caused by *U. diversum* interferes with processes such as artificial insemination and in vitro fertilisation [28,29].

Sequencing and comparative genomic and proteomic analyses of *U. diversum* ATCC 49782 revealed that the *U. diversum* genome comprises 973,501 bp [5,30,31]. The coding sequences (CDS) of antigen genes, haemolysin and the MIB–MIP system (MIB: mycoplasma immunoglobulin (Ig)-binding protein; MIP: mycoplasma Ig protease), involved in pathogenicity have been identified [5,6]. MIB is an IgG-binding protein, whereas MIP cleaves the IgG heavy chain. In addition, several other CDS of potentially immunogenic molecules, including lipid-associated membrane proteins (LAMPs), variable membrane surface lipoproteins (VSPs), and multiple-banded antigens (MBAs), have been identified [6,30,32]. The roles of most of these immunogenic molecules have not yet been investigated, and identifying their functions may help elucidate the pathogenesis of *Ureaplasma* infections. Thirty-seven LAMPs have been described in *U. diversum*, and this protein group has been extensively studied in Mollicutes [33,34,35]. Heterologous expression of LAMPs derived from *Mycoplasma hyopneumoniae* [36], *M. agalactiae* [37], *M. bovis* [38], and other Mollicutes species has led to the development of vaccines and immunodiagnostic tests.

Our previous in silico analysis predicted that the cloning and heterologous expression of certain *U. diversum* LAMPs (including GUDIV-103) are feasible owing to the reduced number of transmembrane loops, good solubility, and low similarity with bovine proteomes and proteins from other Mollicutes that infect cattle [30]. GUDIV-103 showed conformational and linear B-cell epitopes and the potential to bind to different bovine leukocyte antigens (BoLa-1*02301, BoLa-3*00201, BoLa-2*01201, BoLa-6*01301, BoLa-3*00101, BoLa-6*04101, BoLa-1*02301, BoLa-T2C, and BoLa-T5) and was also predicted to be an antigen by the predictor Vaxijen. Other parameters analysed were the instability index, aliphatic index, and hydropathy average [31]. Analyses of all of these parameters indicated that GUDIV-103 is a promising protein for studies on immunogenicity and immunomodulation and a potential target for biotechnological applications because it presents epitopes for B and T (CD8) lymphocytes [4,29,30]. The main objectives of this study were the purification of *U. diversum* GUDIV-103 and the determination of the heterologous expression of GUDIV-103 in *Escherichia coli*. Next, we evaluated the immunogenicity and antigenicity of recombinant GUDIV-103 (rGUDIV-103) and the immunomodulatory effects of rGUDIV-103 in bovine peripheral blood mononuclear cell (PBMC) cultures.

## 2. Materials and Methods

### 2.1. Phylogenetic Analysis

#### 2.1.1. Culture Conditions for *U. diversum* and DNA Extraction

The *U. diversum* ATCC 49782 strain was isolated from a cow with granular vulvovaginitis in Ontario, Canada [39], and 45 *U. diversum* isolates (Appendix A) were provided by the Mycoplasma Laboratory of the Institute of Biomedical Sciences, University of São Paulo, Brazil (USP). Some strains were isolated from cows with granulomatous vulvovaginitis, whereas others were isolated from the semen of healthy bulls. The isolates were obtained from the following four states: 19 from São Paulo, two from Mato Grosso do Sul, one from Minas Gerais, and 22 from Bahia. Next, 1 mL of each sample, previously stored in *Ureaplasma* medium (UB) containing 21 g of Difco PPLO broth, 2 mL of 1% phenol red dye, 20 mL of unheated horse serum ((Invitrogen^®^, Waltham, MA, USA), 10 mL of 25% fresh yeast extract, 1 mL of 10% urea solution (Sigma^®^, St. Louis, MO, USA), and 200,000 units/mL of penicillin G (Sigma^®^) per litre of the medium, was grown in 9 mL of the same medium at 37 °C for 24–48 h [6,29]. Bacterial DNA was extracted using a NucleoSpin kit (Macherey-Nagel, Düren, Germany), according to the manufacturer’s instructions.

#### 2.1.2. Primer Designing, PCR, and Electrophoresis

Previously designed primers (forward: 5′-GTACCTAATCTCAATCAAGC-3′ and reverse: 5′-CAACTAAGTCAACACGAGC-3′) were used for PCR amplification of *GUDIV-103* isolated from *U. diversum* ATCC 49782 [30] and to determine the presence of *GUDIV-103* in the other 45 *U. diversum* strains (Appendix A). Amplifications were performed in a 75-µL mix containing 3 µL DNA, 1 × PCR buffer (10 mM Tris-HCl, pH 9.0; 50 mM KCl), 1.5 mM MgCl_2_, 200 μM dNTP, 50 pmol of each primer, and 1.5 U Taq DNA polymerase (Invitrogen, São Paulo, Brazil) in a Veriti 96-well thermal cycler (Applied Biosystems, São Paulo, Brazil). PCR conditions involved an initial denaturation at 94 °C for 5 min, followed by 35 thermal cycles at 94 °C for 30 s, 54 °C for 30 s, and 72 °C for 1 min, with a final extension at 72 °C for 5 min. The reaction products were analysed using electrophoresis on a 1.5% agarose gel (50 mL), stained with 2.5 µL of 10 mg/mL ethidium bromide, and visualised and photographed under UV light using a MultiDoc-It Imaging System (UVP, Analytik Jena). A 1 kb Plus DNA ladder (Invitrogen) was used to assess the amplified fragment size.

#### 2.1.3. PCR Product Purification, Sequencing, and Phylogenetic Tree Construction

After electrophoresis, the remaining PCR mixture (63 μL) was precipitated with 500 μL 65% isopropanol. The sample was centrifuged at 19,000× *g* for 5 min, washed with 250 μL of 70% alcohol, and re-centrifuged for 5 min. The remaining solvent was removed by placing the tubes in an inverted position for 1 h at room temperature (24 °C). The DNA pellet was suspended in 20 μL of ultrapure water. Sanger sequencing was performed at the Institute of Biomedical Sciences of the Federal University of São Paulo according to the MegaBACE 1000 protocol, using the DYEnamic ET Dye Terminator kit on DNA Polymerase Thermo Sequenase II (Amersham Biosciences, US81090, Amersham, UK). The obtained sequences were processed using sequence analyser software with the Cimarron Caller Base 3.12. *GUDIV-103* sequences were obtained as chromatograms and compared with those submitted to GenBank using BLAST (National Center for Biotechnology Information, NCBI; accession number: NZ_CP009770). Sequence alignment was performed using MEGA- X version 4.1 [40,41] by Clustal with a bootstrap value of 1000, and the tree was built using the neighbour-joining method with a Tajima–Nei distance matrix.

### 2.2. Purification of rGUDIV-103 and Determination of Its Expression

#### 2.2.1. Cloning, Protein Expression, and Solubility Tests

The full-length sequence of GUDIV-103 isolated from *U. diversum* ATCC 49782 was synthesised using GenScript (Piscataway, NJ, USA) and cloned between the *Nde*I and *Xho*I sites of the pET-28a(+) expression vector to generate N-his6-tagged-GUDIV-103 using standard molecular techniques. The vector contained a kanamycin-resistance cassette. The expression plasmid was transformed into One Shot TOP10 competent cells (Thermo Fisher Scientific, Waltham, MA, USA), selected on Luria-Bertani (LB) medium (with 1.5% agarose) supplemented with 100 µg/mL kanamycin, and cultured overnight at 37 °C. Individual colonies were cultured overnight in an orbital shaker at 37 °C and 200 rpm. Next, 1 mL aliquots of two chosen colonies were used for plasmid purification using the QIAprep spin miniprep kit (Qiagen, Hilden, Germany) and confirmed by electrophoretic separation on 1.5% agarose gel. The isolated pET-28a(+) expression vector was transformed into *E. coli* One Shot BL21 Star (DE3) cells (Invitrogen) for protein expression. The plasmids were extracted, and the presence of the *GUDIV-103* gene was confirmed using PCR.

Following the heterologous expression, rGUDIV-103 solubility was tested as follows: 50 mL of LB medium was inoculated with 500 μL *E. coli* BL21 (DE3) culture and incubated at 37 °C with shaking to an optical density (OD_600 nm_) not greater than 0.600. A 1 mL aliquot of this culture was centrifuged at 10,000× *g* for 10 min. The pellet was homogenised with 100 µL 0.1 M Tris buffer, 0.5 M NaCl, and 10% glycerol at pH 8.5, and a 12 µL aliquot was used as a time zero representative. Isopropyl β-d-1-thiogalactopyranoside was added to the remaining culture (1 mM final concentration) and grown overnight at 18 °C on a shaker. An aliquot (200 μL) was removed and pelleted under the same conditions to prepare the overnight sample. The remaining culture was centrifuged, and the pellet was homogenised with 0.1 M Tris-HCl, 0.5 M NaCl, and 10% glycerol at pH 8.5. The cells were lysed by sonication for 90 s (10 s on, 15 s off) at 35% amplitude (Digital Sonifier Cell Disruptor, Branson Ultrasonics, VWR Scientific, Canada). The lysates were centrifuged; the supernatant represented the soluble fraction, and the pellet corresponded to the insoluble fraction. The pellet was homogenised in 5 mL of suitable buffer (0.1 M Tris buffer, 0.5 M NaCl, 10% glycerol, pH 8.5 buffer) supplemented with 8 M urea. The presence of histidine-tagged proteins was determined using sodium dodecyl sulphate–polyacrylamide gel electrophoresis (SDS–PAGE), followed by Western blotting.

#### 2.2.2. SDS–PAGE and Western Blotting

*E. coli* BL21 (DE3) rGUDIV-103 cell fractions (20 µL) were homogenised in 20 µL SDS–PAGE sample buffer (0.15 M Tris-HCl pH 6.8, 10% mercaptoethanol, 1.2% SDS, 30% glycerol, 0.04% bromophenol blue) and boiled for 5 min. The samples were loaded onto a 12% SDS polyacrylamide gel (0.5 M Tris-HCl pH 6.8; 0.1% SDS; 12%/0.8% acrylamide/bisacrylamide; 0.05% ammonium persulfate; 0.05% N, N, N’, 50 N’-tetramethylenediamine) and separated by electrophoresis in a running buffer (25 mM Tris-HCl pH 8.8; 250 mM glycine; 0.1% SDS) at 120 mV for 60 min. The gel was stained with Coomassie blue solution (10% acetic acid, 40% methanol, and 1% Coomassie Brilliant Blue G-250) to detect protein bands. Western blotting was performed by transferring the proteins from the unstained gel to a 0.45 µm nitrocellulose membrane (Bio-Rad) with a Mini Trans-Blot Cell blotter (Bio-Rad, Hercules, CA, USA) at 120 V and 250–400 mA for 90 min using transfer buffer (25 mM Tris-HCl at pH 8.3, 192 mM glycine, 20% methanol). The membrane was incubated in PBS-T (50 mM sodium phosphate, 150 mM sodium chloride, 0.05% Tween 20) and 3% skim milk overnight at 4 °C, washed thrice (5 min each) with PBS-T, and incubated for 2 h at room temperature with 1:3000 6x-His epitope tag primary antibody (Invitrogen) in PBS-T/skim milk. The membrane was washed and incubated for 2 h with horseradish peroxidase (HRP)-conjugated anti-rabbit IgG (Invitrogen) diluted to 1:10,000 in PBS-T. The membrane was developed using 3, 3′, 5, 5′-tetramethylbenzidine (TMB) according to the manufacturer’s instructions (Sigma-Aldrich, St. Louis, MO, USA).

#### 2.2.3. Protein Extraction and Purification

*E. coli* was cultured in 1 L LB medium under the conditions described above. The culture was centrifuged at 10,000× *g* for 10 min at 4 °C, and the pellet was homogenised in 200 mL of 0.1 M Tris-HCl at pH 8.5, 0.5 M NaCl, 10% glycerol, and 1 mM phenylmethylsulphonyl fluoride (PMSF). The bacterial cells were mechanically lysed in an APLAB-10 homogeniser (ARTEPEÇAS, São Paulo, Brazil) at 600 psi for 15 min, centrifuged at 14,000× *g* for 1 h at 4 °C, and the supernatant was filtered through a membrane with 0.22 μm pore size.

The supernatant was applied to a nickel-chelating resin (1 mL HisTrap HP; GE Healthcare Biosciences Corp., UK) using a peristaltic pump (Watson Marlow sci 323) pre-equilibrated with binding buffer (0.1 M Tris-HCl at pH 8.5, 0.5 M NaCl, and 10% glycerol). The resin was washed and eluted with a binding buffer containing 25–500 mM imidazole. Purification was followed by spectrophotometry at 280 nm using a dual-beam spectrophotometer (Kazuaki), and protein bands were visualised by SDS–PAGE and Western blotting. Fractions containing purified protein were dialysed in 0.1 M Tris-HCl of pH 8.5, 0.5 M NaCl, and 10% glycerol overnight at 4 °C using SnakeSkin dialysis tubing (3.5 K MWCO, 35 mm; Thermo Scientific). Protein concentrations were determined using the Bradford assay. Bradford’s reagent was prepared by diluting 100 mg of Coomassie Brilliant Blue BG-250 in 50 mL ethanol. Subsequently, 100 mL of 85% (*w*/*v*) phosphoric acid was added, the mixture diluted to 1 L, and stored at 4 °C. For protein determination, 10 µL of Bradford reagent was added to 10 µL of the protein solution. The absorbance of the solution was measured at 595 nm using an iMark microplate absorbance reader (Bio-Rad). The protein concentration was determined using a calibration curve based on bovine serum albumin as a standard.

### 2.3. Antigenicity and Immunogenicity Assays

#### 2.3.1. Production of Polyclonal Antibodies

The animal experiments were approved by the Animal Use Ethics Committee (CEUA), Faculty of Medicine, University of São Paulo (FMUSP), and the Federal University of Bahia, Multidisciplinary Institute in Health, Campus Anísio Teixeira (process number: FMUSP—1201/2018; CEUA-IMS/CAT-UFBA—048/2017). Two female New Zealand rabbits were kept in individual cages and provided with food and water ad libitum. Blood samples were obtained from the marginal ear vein prior to inoculation, to be used as a negative control (pre-immune). Immunisation was performed intramuscularly in the quadriceps region on the posterior surface of the right and left thighs with 500 μg of recombinant protein mixed with an equal volume of Freund’s complete adjuvant (Sigma-Aldrich). Two subsequent immunisations were performed at two-week intervals using 500 μg of recombinant protein mixed with Freund’s incomplete adjuvant. Post-immunisation blood was collected every seven days, and the obtained serum samples were stored at −80 °C. On the 42nd day after immunisation, animals were anaesthetised by intramuscular injection of 30–40 mg/kg ketamine and 5–10 mg/kg xylazine, followed by cardiac puncture exsanguination. The animals were euthanised by intravenous administration of 40–50 mg/kg thiopental sodium.

#### 2.3.2. Purification of Serum IgG Fractions

IgG antibody fractions against rGUDIV-103 were delipidated and dialysed [37], followed by purification using a HiTrap protein G HP affinity column (GE Healthcare Biosciences, Piscataway, NJ, USA), eluted in 0.1 M glycine (pH 2.7), neutralised with 1 M Tris-HCl pH 9.0, and dialysed in PBS of pH 7.4. IgG antibody fractions were monitored using 12.5% SDS–PAGE. The purified antibodies were used for Western blotting, enzyme-linked immunosorbent assay (ELISA), and dot blot assay.

#### 2.3.3. Indirect ELISA

Polystyrene microplates (Corning COSTAR) were coated with rGUDIV-103 (200 ng/well) diluted in 0.5 M carbonate–bicarbonate buffer (pH 9.6) for 16 h at 4 °C in a humidity chamber and washed thrice with PBS-T. Non-specific binding was blocked with 10% skimmed milk in PBS-T for 2 h at 37 °C and washing thrice with PBS-T. Serum from rabbits immunised with recombinant protein or *U. diversum* ATCC 49783 (anti-*Ud* serum) was added at different dilutions, and the microplates were incubated for 2 h at room temperature. The plates were washed three times with PBS-T and incubated for 1 h and 30 min with peroxidase-conjugated secondary antibody (HRP-conjugated goat anti-rabbit IgG, Invitrogen) diluted at 1:2000 in PBS containing 5% skim milk. The wells were washed three times with 200 μL PBS-T and developed with 100 μL TMB Single Solution (Novex, Life Technologies) for 15 min. The reaction was stopped by adding 50 μL of 1 N hydrochloric acid, and the absorbance was read at 450 nm using a microplate reader.

#### 2.3.4. Kinetics of Mono-Specific Polyclonal Antibodies

Indirect ELISA was used to evaluate the immunisation kinetics (anti-rGUDIV-103 titres). In-house indirect ELISA was performed after validation, according to Barbosa et al. (2020). The plates were coated with rGUDIV-103 (200 ng/well), followed by the addition of different dilutions of rabbit sera containing antibodies against rGUDIV-103, and incubated for 1 h with a secondary antibody (HRP-labelled goat anti-rabbit IgG H + L, 1: 800). Samples and background controls (reactions with or without the primary antibody) were evaluated in triplicate using biological and technical samples.

#### 2.3.5. Dot Blotting

A 5 μL solution containing 30% (*w*/*v*) skim milk in PBS-T and 5 μg of rGUDIV-103 was applied to a 0.45 µm nitrocellulose membrane (Bio-Rad), allowed to dry, blocked with 30% (*w*/*v*) skim milk in PBS-T, and incubated at 4 °C overnight. The membranes were washed three times with PBS-T and incubated for 2 h with pre-immune and hyper-immune serum produced against rGUDIV-103 or serum produced against *U. diversum* ATCC 49783 (1:200 in PBS-T). Incubation with PBS-T alone served as a negative control. The membranes were washed five times with PBS-T and incubated for 2 h with HRP-conjugated goat anti-rabbit IgG (Invitrogen) at a 1:10,000 dilution under agitation. After five washes, the bound protein was visualised by staining with 3, 3′-diaminobenzidine (DAB) substrate for 5 min. The reaction was stopped by rinsing the membrane with distilled water [42].

#### 2.3.6. LAMP Extraction and Western Blotting

LAMPs were extracted using the methodology described by Marques et al. [6]. As per this method, 100 mL of the *U. diversum* culture, colour changing units of approximately 10^5^ cells/mL, from 42 strains (out of 45) were harvested by centrifuging at 22,700× *g* for 30 min at 4 °C. The pellet was washed twice with PBS) for 15 min at 4 °C. Proteins were separated into hydrophobic and hydrophilic fractions using the Triton X-114 partition method [43], separated using 12.5% SDS–PAGE, transferred to a nitrocellulose membrane, and analysed by Western blotting using polyclonal anti-rGUDIV-103 antibodies. The results were compared with those of PCR amplification using *GUDIV-10*3 genes [30].

### 2.4. Immunomodulation

#### 2.4.1. Bovine Peripheral Blood Collection and PBMC Isolation

Blood samples were collected from adult Holstein cows in a rural area of Bahia, Brazil. Bovine peripheral blood was collected from the jugular vein into vacutainer tubes containing 1.5 mg/mL EDTA (BD Vacutainer^®^) and diluted at 1:1 in PBS (pH 7.4). Next, 10 mL of the mixture was added to 3 mL of Ficoll–Histopaque solution (density: 1.0771 g/mL, Sigma-Aldrich, São Paulo, Brazil) in a 15 mL tube to produce the Ficoll–Histopaque barrier, followed by centrifugation at 4200× *g* for 20 min at room temperature. The mononuclear cells present at the Ficoll/plasma interface were removed, washed twice in PBS, and centrifuged at 4200× *g* for 10 min after each wash step. PBMCs were counted in a Neubauer chamber, and cell viability was assessed using 0.1% trypan blue (viability > 90%). Viable cell concentrations were adjusted to 1 × 10^6^ cells/mL by adding Roswell Park Memorial Institute (RPMI)-1640 medium supplemented with 10% foetal bovine serum (FBS; Gibco, São Paulo, Brazil) and cultured in complete RPMI-1640 medium supplemented with 10% FBS and 100 U/mL penicillin.

#### 2.4.2. rGUDIV-103 Incubation with Bovine Cell Cultures

Lipopolysaccharide (LPS) was removed from PBMC using the protocol described by Reichelt et al. [44]. According to this protocol, 1 × 10^6^ cells/mL were incubated with 0.5, 1.0, 2.0 and 4.0 μg/mL of rGUDIV-103 dialysed in PBS (pH 7.4) for 2 or 6 h. The cells were inoculated in PBS (7.4) and 100 ng/mL LPS, which were the negative and positive controls, respectively. After incubation, the cells were resuspended in RNAlater (Thermo Fisher Scientific, Waltham, MA, USA), and the supernatant was collected and frozen at −70 °C for mRNA extraction and cDNA synthesis.

#### 2.4.3. Proliferation Assay

Cell proliferation was determined using 5(6)-carboxyfluorescein diacetate succinimidyl ester (CFSE; Invitrogen) according to the manufacturer’s specifications. Firstly, the cells were counted in a Neubauer chamber with 0.1% trypan blue and incubated with CFSE for 20 min at 37 °C and 5% CO_2._ The cells were then centrifuged at 200× *g* for 10 min and resuspended in complete RPMI-1640 medium. The cell count was adjusted to 1 × 10^6^ cells/mL, dispensed in triplicate to a 24-well COSTAR cell culture plate (Corning), and incubated with 0.5, 1.0, 2.0 and 4.0 μg/mL rGUDIV-103 for 60 h. PBS (pH 7.4) and 5.0 μg/mL concanavalin A (ConA, Sigma-Aldrich) were used as negative and positive controls, respectively.

#### 2.4.4. Gene Expression

RNA was extracted using the PicoPure RNA isolation kit with DNase treatment (Thermo Fisher Scientific), and RNA elution was performed in 11 μL of the eluting solution. cDNA was obtained from the mRNA using a SuperScript IV reverse transcriptase kit (Thermo Fisher Scientific). Specific primers for interleukin 1 beta (IL-1β) [45], tumour necrosis factor alpha (TNF-α) [45], toll-like receptor 2 (TLR2) [45], TLR4 [45], inducible nitric oxide synthase (iNOS) [46], caspase-3 (Casp3) [47], and glyceraldehyde 3-phosphate dehydrogenase (GAPDH) [45] were used in quantitative PCR to determine gene expression. The reaction was performed using a StepOne Real-Time PCR System (Applied Biosystems, São Paulo, Brazil). The melting curve was evaluated at the end of the reaction to determine the specificity of the amplification. Gene expression was analysed using the 2^−ΔΔCT^ method [48]. GAPDH was used as an endogenous gene to evaluate the overall cDNA content.

#### 2.4.5. H_2_O_2_ and NO Quantification

Hydrogen peroxide (H_2_O_2_) levels in PBMC and PMN cultures were measured using the Amplex Red hydrogen peroxide/peroxidase kit, according to the manufacturer’s recommendations (Thermo Fisher Scientific). After preparing the kit stock solutions, 50 µL aliquots of the standard curve samples, controls, and experimental samples were added to individual wells of a 96-well microplate (Greiner Bio-One, Kremsmünster, Austria). The Amplex Red reagent/HRP working solution (50 μL) was added to the wells and incubated for 30 min at room temperature (~25 °C) in darkness. The absorbance at 550 nm was measured using a microplate reader, and the H_2_O_2_ concentration (μM) of the PBMC culture supernatant was determined using a standard curve. Nitric oxide (NO) levels were quantified using the Griess assay [49]. Briefly, 50 μL aliquots of the test solution (standard, control, experimental sample) were added to 0.1% Griess reagent solution (50 μL of 3.9 mM N-(1-naphthyl)ethylenediamine in 5% (*v*/*v*) phosphoric acid) and incubated in the dark at room temperature for 10 min. Sulphanilamide solution (1% in phosphoric acid) was added to the mixture, and the absorbance of the coloured product was measured at 540 nm using a microplate reader. Standard sodium nitrate solutions (0.1 mM) were used to construct a standard curve to determine the actual concentrations of the samples. The same experiment was performed with total lipoproteins extracted from strain ATCC 49782, as well as with the aforementioned viable and heat-inactivated strain (70 °C, 30 min).

### 2.5. Statistical Analysis

Normality was assessed using the Shapiro–Wilk normality test. One-way analysis of variance with Bonferroni test was used for normally distributed data. The non-parametric Kruskal–Wallis test, followed by Dunn’s post hoc test, was used where the data did not fulfil the criteria of normality. Results were expressed as the mean ± standard error of the mean. Statistical significance was set (*p*) < 0.05. All experiments were repeated thrice using independent biological replicates, and statistical analyses were performed using GraphPad Prism version 5.0.

## 3. Results

### 3.1. Cloning GUDIV-103 Inserted into pET-28a(+) Expression Vector in E. coli BL21 Star (DE3)

The 732-nucleotide sequence inserted into the pET-28a(+) vector contained 663 *GUDIV-103*–encoded nucleotides, with the remaining nucleotides representing stop codons (TGA replaced by TGG), a start codon, restriction enzyme cleavage sites, and the histidine tail (Figure 1A). After transformation, the 6000 bp vector from the *E. coli* One Shot TOP10 transformants was confirmed on an agarose gel (Figure 1B). A ~300 bp fragment from the *GUDIV-103* coding sequence was confirmed in *E. coli* BL21 (DE3) via PCR after DNA extraction (Figure 1C).

### 3.2. Expression and Purification of Soluble rGUDIV-103 from E. coli BL21 (DE3) Star

Minimal expression was observed in the absence of induction (Figure 1D, T-zero). rGUDIV-103 was expressed in the soluble and insoluble fractions at 18 °C, whereas only insoluble proteins were observed at 37 °C by SDS polyacrylamide gel and Western blotting (Figure 1D). The molecular weight of rGUDIV-103 was consistent with the theoretical value (27.32 kDa) predicted by the ProtParam server (https://web.expasy.org/protparam/ (accessed on 19 June 2021)). Therefore, rGUDIV-103 was subsequently expressed at 18 °C.

Ni^2+^-affinity purification of N-his_6_-tagged rGUDIV-103 involved stepwise increase in the imidazole concentration. Approximately 5 mL of binding buffer and 25, 120, and 500 mM imidazole were successively passed through a 1 mL HisTrap column with stepwise increase in the imidazole concentration when A280 nm readings were zero (Figure 1E). After elution with 500 mM imidazole, multiple protein bands were observed, indicating that the intermediate eluent concentration (120 mM imidazole) was insufficient to remove all the impurities. The replacement of 120 mM imidazole with 160 mM imidazole resulted in the highest rGUDIV-103 purity (Figure 1F).

### 3.3. rGUDIV-103 Is an Immunogenic and Antigenic Protein

Production of specific IgG antibodies against rGUDIV-103 significantly increased (*p* < 0.05) in rabbits inoculated with rGUDIV-103 in a time- and dose-dependent manner (Figure 2A).

Antibodies produced against rGUDIV-103 and *U. diversum* ATCC 49783 were purified (Figure 2B,C) and used for indirect ELISA and dot blot assays (Figure 2C,D). Indirect ELISA showed that the content of purified and non-purified anti-*Ud* strain ATCC 49783 IgG was higher than that of the pre-immune antibodies (*p* < 0.05). The immunoreactive response of the purified anti-rGUDIV-103 antibody was significantly greater than that of the pre-immune and anti-*Ud* antibody (*p* < 0.05; Figure 2C). Dot blot assay showed that anti-*Ud* antibodies bound to rGUDIV-103, and no reactivity was observed against the pre-immune serum or negative control (Figure 2D).

### 3.4. Intraspecific Analysis and Phylogenetic Tree Construction Revealed GUDIV-103 Polymorphism and Genetic Proximity among Strains from the Same Locality

*GUDIV-103* was found in 30 of the 46 *U. diversum* strains isolated from different Brazilian states (Bahia, Mato Grosso do Sul, Minas Gerais, and São Paulo) using PCR (Table 1 and Figure 3A). Alignment of *GUDIV-103* from the reference strain *U. diversum* ATCC 49782 (GenBank: CP009770) with *GUDIV-103* sequences from the isolated strains revealed polymorphisms, insertions, and deletions in some regions. The phylogenetic tree (Figure 3B) revealed genetic proximity between isolates collected from the same state and farm (Table 1 and Appendix A). The first cluster comprised samples collected from a property located in Mato Grosso do Sul (Figure 3B; strains 9653 and 805 are underlined in blue). The samples collected in Bahia (underlined in green) and São Paulo (underlined in black) formed different clusters, whereas isolates from an artificial insemination centre in São Paulo, from where frozen semen samples were obtained (16T, 13T, 12T, 10T, 7T, 5T), were grouped together (Figure 3B).

### 3.5. GUDIV-103 Was Detected in the Total Membrane Protein Extract of U. diversum Strains

In total, 30 of *46 U. diversum* strains were PCR positive for *GUDIV**-103* (Figure 3A, Table 1). Total LAMPs were extracted from 42 strains; the growth of the remaining four strains was not sufficient for extraction. GUDIV-103 was only observed in the hydrophobic phase following TX-114 extraction using Western blotting with anti-rGUDIV-103 antibodies (Table 1).

### 3.6. rGUDIV-103 Induced NO and H_2_O_2_ Production in Bovine PBMCs

NO production significantly increased in bovine PBMC treated with 0.5, 1.0, 2.0, and 4.0 µg/mL rGUDIV-103 for 2 h compared with that of the negative control (NC; Figure 4A). A significant increase in NO production was also observed on inducing bovine PBMC with the three lowest rGUDIV-103 concentrations for 6 h (Figure 4B). An increase in NO production was also observed when cell cultures were treated with viable and inactive *U. diversum*, as well as when treated with different concentrations of total lipoproteins extracted from strain ATCC 49782 (Appendix A).

Furthermore, H_2_O_2_ production significantly increased in PBMC treated with 4.0 µg/mL rGUDIV-103 for 2 h (*p* < 0.05) compared with that in the NC (Figure 4C) and significantly increased after 1.0, 2.0, and 4.0 µg/mL of rGUDIV-103 treatment for 6 h (Figure 4D).

iNOS transcript levels significantly increased (*p* < 0.05) in PBMC treated with all four rGUDIV-103 concentrations for 2 h compared with that in the NC (Figure 4E) and after 2.0 and 4.0 µg/mL rGUDIV-103 treatment for 6 h (Figure 4F).

### 3.7. rGUDIV-103 Induced Pro-Inflammatory Gene Expression in Bovine PBMCs

*IL-1β*, *TNF-α*, *TLR2*, and *TLR4* (Figure 5A–D) expressions increased more (*p* < 0.05) in PBMC treated with 0.5, 1.0, 2.0, and 4.0 µg/mL rGUDIV-103 for 2 h than that in the NC. Statistical differences in expression were observed at certain instances, particularly in *TNF-α* after 2.0 µg/mL rGUDIV-103 treatment (Figure 5B) and *TLR4* after 4.0 µg/mL rGUDIV-103 treatment (Figure 5D).

*IL-1β* expression increased significantly (*p* < 0.05) in PBMC cultures treated with all concentrations of rGUDIV-103 for 6 h compared with that in the NC (Figure 5E). *TNF-α* (Figure 5F) and *TLR4* (Figure 5H) expression was significantly altered (*p* < 0.05) upon treatment with 1.0, 2.0, and 4.0 µg/mL rGUDIV-103, whereas *TLR2* expression (Figure 5G) in PBMC treated with 2.0 and 4.0 µg/mL rGUDIV-103 was different from that in the NC (*p* < 0.05). Moreover, *IL-1β* expression in PBMC treated with 1.0 µg/mL rGUDIV-103 (Figure 5E) and *TNF-α*, *TLR2*, and *TLR4* expressions in PBMC treated with 4.0 µg/mL rGUDIV-103 were also statistically different from those in the NC.

Lymphoproliferation induction in bovine PBMCs treated with 0.5 µg/mL rGUDIV-103 was statistically significant (*p* < 0.05, Figure 6A). However, the proliferation of PBMC treated with 1.0, 2.0, and 4.0 µg/mL rGUDIV-103 was insignificant compared with that of the NC (Figure 6A).

The expression of the pro-apoptotic caspase-3 gene in PBMCs treated with 1.0, 2.0, and 4.0 µg/mL rGUDIV-103 for 2 h was not significantly different from that seen in the NC group (data not shown). However, caspase-3 expression increased in a dose-dependent manner after incubation for 6 h (Figure 7).

## 4. Discussion

Initial studies on bovine *Ureaplasma* species indicated that these microorganisms infect the genital, reproductive, and respiratory tracts [18,50,51,52]. Studies on these bacteria have highlighted the specific molecules involved in their pathology. For example, the virulence factors associated with *U. diversum* pathogenicity, which cause reproductive disorders in cows and bulls, were elucidated via sequencing and genomic analysis in 2015 and 2016, respectively [5,6]. The *U. diversum* genome contains several CDS of potentially immunogenic molecules, including LAMPs, VSPs, MBA, haemolysins, and the MIB–MIP system [6]. *U. diversum* encodes 37 CDS of LAMPs, which are considered the main virulence factors of Mollicutes [5,6]. Previous in silico analyses predicted that some LAMPs, such as GUDIV-103, may stimulate humoral and cellular responses and have favourable properties for cloning and heterologous expression [30]. Santos-Junior et al. [30] suggested that rGUDIV-103 overexpression in *E. coli* might produce soluble proteins using protein-sol and SOLpro predictors based on its physicochemical characteristics: predicted molecular weight of 27 kDa, instability index below 40, presence of a signalling peptide for excretion (by classical and non-classical pathways), and negative grand average of hydropathicity. Despite the advantages of using *E. coli* as an efficient heterologous expression system, insoluble proteins may still be produced [53,54,55]. Soluble expression is highly desirable, as solubilisation of insoluble proteins involves the use of strong denaturants and requires refolding, which is rarely successful and often leads to the loss of biological activity [56,57,58]. In this study, the pET-28a(+) vector harboured the *GUDIV-103* gene in the *E. coli* BL21 (DE3) Star expression system, as confirmed using PCR, and yielded a large-scale heterologous expression of soluble rGUDIV-103 at 18 °C. This result fulfilled the in silico prediction [30] and eliminated the need for additional solubilisation and refolding.

Initial studies on *U. diversum* indicated that it induces a humoral response in the host [50,59]. Antisera are produced in rabbits or calves by inoculating viable *U. diversum* to diagnose or inhibit in vitro *U. diversum* growth [7,50,59,60,61]. Several studies have highlighted the immunogenic potential of LAMPs derived from Mollicutes. Recombinant DNA technology has enhanced our knowledge of the structure and immunobiology of these biomolecules. In this study, rabbit immunisation with rGUDIV-103 increased IgG titres over 42 days, as detected using direct ELISA. Purified or impurified rGUDIV-103- or ATCC 49783-immunised rabbit sera were specific for rGUDIV-103. This result strongly suggested that rGUDIV-103 played an important role in generating a humoral response in the host by producing specific antibodies. Recombinant proteins from other Mollicutes, such as Mhp597 from *M. hyopneumoniae* [62], MbovP280 from *M. bovis* [63], surface proteins (MAG_1560, MAG_6130, and P40) from *M. agalactiae* [37], and GrpE from *U. urealyticum* [64], also induce antibody production.

Serum (antibody) production against *U. diversum* involves selecting specific strains to represent the entire repertoire of isolates worldwide [59,60], with some sera classified into three clusters (A, B, or C) [65,66]. However, this classification is indicative of the relationship between selected strains, and not the complete profile of antigens in *U. diversum*, owing to the large variability in the isolates based on the serological relationship of antigens from bovine *Ureaplasma*. Our data showed considerable intraspecific genetic variability among *GUDIV-103* isolates from different Brazilian states. However, isolates from the same localities (state or farm) had low gene variability as they were grouped in the same clade or closely related clades in the phylogenetic tree. Lipoprotein variation in Mollicutes contributes to host immune evasion [67,68]. However, Mollicutes also uses antigenic variation as an adaptation mechanism [69]. *M. hominis* showed no antigenic variation over time when evaluated in the same individual; in contrast, a high degree of variation was observed in different patients [70]. Therefore, antigenic variation is associated with the adaptation of the strain to escape the response elicited by the host, although mechanisms associated with the adaptation to a specific host physiology cannot be ruled out.

In this study, *GUDIV-103* was detected in 30 of 46 *U. diversum* strains using PCR. However, all samples were detected positive for GUDIV-103 using Western blotting with sera containing anti-rGUDIV-103 antibodies. This result strongly suggested significant variation in the *GUDIV-103* sequence. Positive Western blotting results, even for PCR-negative strains, may be related to gene alterations that do not lead to substantial changes in the primary structure of the lipoproteins. Marques et al. [71] verified the intraspecific variability and single nucleotide polymorphisms in 16S rRNA gene fragment sequences from *Ureaplasma* derived from healthy and unhealthy cattle. The mechanism involved in the antigenic variation in *U. diversum* is poorly explored; however, variations in *Ureaplasma* spp. MBA/*mba* in human placentae demonstrated that this variation is associated with the incidence and severity of chorioamnionitis during pregnancy [72]. Many *U. diversum* strains used in this study were isolated from cows with reproductive disorders; therefore, the mechanism of antigenic variation of GUDIV-103 lipoprotein, as well as the reason why strains from different clades (such as 805 and ATCC) are amplified by PCR using the same pair of primers, should be further investigated. This variability may translate into antigenic diversity among different strains, resulting in different types of interactions with the host [29,30,71,73].

Our data revealed that bovine PBMC cultures incubated with rGUDIV-103 for 2 and 6 h induced NO and H_2_O_2_ production and iNOS expression. Many other Mollicutes, LAMP extracts, and recombinant LAMPs induce NO and H_2_O_2_ production in cellular models [74,75]. NO is a multifunctional molecule involved in several physiological and pathological processes and often plays a dual concentration-dependent role. *U. diversum* induces NO production in peritoneal macrophage cultures, associated with a pro-inflammatory cytokine profile [76]. On inoculation with *U. diversum*, cells from the bovine endometrium, vagina, and peripheral blood produce NO, suggesting that these cells play a role in the immediate response to *Ureaplasma* infection [75]. Low NO concentrations protect cells from apoptosis, whereas excessive NO production triggers apoptosis in several cell types [74,77]. The induction of NO production is widely associated with the induction of apoptosis in Mollicute-infected cells. Many *Mycoplasma* spp. trigger apoptosis by inducing excessive NO production [74,78,79]. Reactive oxygen species (ROS) are implicated in the apoptotic cascade and trigger the activation of initiator and executioner caspases [80]. Choi et al. [81] showed that LAMP from *M. pneumoniae* elevates ROS levels in A549 lung epithelial cells, which is associated with apoptosis in LAMP-exposed cells [78]. Other studies showed NO induction and ROS production in cells inoculated with Mollicutes [82,83]. In several studies, LAMPs have been associated with pro-inflammatory cytokine production or apoptosis [29,84].

In the present study, caspase-3 gene expression increased in a dose-dependent manner in PBMC cultures on incubation with rGUDIV-103. Although we did not directly assess apoptosis in rGUDIV-103–treated cells, increases in caspase activation, ROS production, and NO production are important molecular events in the apoptotic cascade [80,83]. Excessive ROS levels lead to mitochondrial dysfunction, which triggers the activation of caspase cascades and apoptosis [85]. Caspase-3 induces the downstream effector caspases [78,83]. In many animal cells infected with Mollicutes, apoptosis execution phase correlates with caspase-3 activation [78,86,87]. Some *U. diversum* strains induce apoptosis in human epithelial carcinoma (Hep-2) cells; however, the number of apoptotic cells and pro-apoptotic caspase (2, 3, and 9) expression levels decrease over time [88]. This reduction could be associated with the persistence of these microorganisms in the intracellular environment. Therefore, although our study strongly correlated rGUDIV-103 with increased occurrence of apoptosis, other *U. diversum* virulence factors associated with host adaptation, such as cell invasion [1,88] and induction of an anti-inflammatory cytokine cascade [4,73], may suppress apoptosis-related events. In this study, although adequate controls were used for the applied methodologies, a limitation was that a mutant without the lipid-anchor signal was not used in the experiments.

Sonicated *U. diversum* supernatant is mitogenic in murine spleen lymphocytes [76]. In this study, 0.5 µg/mL rGUDIV-103 induced lymphoproliferation in PBMC; however, mitogenic activity decreased and caspase-3 gene expression increased as the rGUDIV-103 concentration increased. This suggested that low rGUDIV-103 concentrations induced an immunogenic response mediated by lymphocyte proliferation; however, high doses may have deleterious effects on cells. The highly conserved antigen nucleotide exchange factor (GrpE) induces an immune reaction against *U. urealyticum* by stimulating a proliferative response in murine lymphocytes [64]. Meanwhile, Mollicutes and their LAMPs modulate the host immune response and contribute to immune escape during acute infection, resulting in chronic diseases [4,68]. Bacteria associated with chronic non-progressive pneumonia in sheep and goats (*M. ovipneumoniae*) induce polyclonal suppression of CD4^+^, CD8^+^, and B lymphocyte subsets in vitro [89]. In this study, rGUDIV-103 was found to stimulate various components of the immune system.

The expression of pro-inflammatory genes (*IL-1β*, *TNF-α*, *TLR2*, and *TLR4*) in bovine PBMCs increased after incubating with 0.5, 1.0, 2.0, and 4.0 µg/mL rGUDIV-103 for 2 h. However, only *TNF-α* and *TLR4* were significantly expressed after incubating with 0.5 µg/mL rGUDIV-103 for 2 h, and *TLR2* was significantly expressed after incubating with 0.5 and 1.0 µg/mL rGUDIV-103 for 6 h. In Mollicutes, cytokine production is associated with the interaction between LAMPs and TLRs [33]. TLR2 and TLR6 play important roles in signal transduction, leading to the production of pro-inflammatory cytokines. Inoculation of viable or heat-inactivated *U. diversum* into murine macrophages increased *TLR2* gene expression associated with the production of IL-1β and TNF-α [6]. *U. diversum* inoculations frequently induce *TNF-α*, *IL-1β*, and *IL-6* expression [31,76]. Pro-inflammatory cytokine stimulation associated with *TLR2* and *TLR6* activation or expression has been defined in several Mollicutes species that infect diverse hosts [35,63,68,74]. Although TLR2 seems to play an important role in the production of pro-inflammatory cytokines, strong evidence indicates that signalling via TLR4 is directly associated with IL-1β and TNF-α expression when macrophages are inoculated with total LAMP extracts, viable *U. diversum*, or heat-inactivated *U. diversum* [4,29]. This pro-inflammatory profile may be linked to the presence of GUDIV-103 in the membranes of these bacteria.

Our data correlated with those of a previous study by Marques et al. [71], indicating marked diversity among *U. diversum* strains. Therefore, as in other Mollicutes [90], variable antigenic profiles cause different interactions with the host immune system. Some *U. diversum* strains result in immunosuppression by significantly inducing IL-10 expression upon incubation with bovine macrophages [73]. This increases the susceptibility of the host to other disease-causing pathogens and interferes with the effectiveness of experimental vaccinations against these pathogens [73,91]. rGUDIV-103 stimulated lymphoproliferation and induced *iNOS*, *IL-1β*, *TNF-α*, *TLR2*, and *TLR4* expression, suggesting that it is a promising target protein for immunobiological studies, as immunostimulation activates the elements capable of fighting infections and cellular and humoral immune responses. However, studies evaluating whether immunological pressure would result in the loss of expression of this protein (with minimal cost of adaptation to the bacteria) should be carried out to authenticate the potential immunobiological application of GUDIV-103. Despite negative PCR amplification of the *GUDIV-103* gene in 16 of the 46 strains, anti-rGUDIV-103 IgG bound to the total LAMP extracts from the 42 viable strains tested. This implied that changes in the *GUDIV-103* sequence of these strains did not sufficiently change the epitope to prevent antigen–antibody interactions.

## 5. Conclusions

Soluble *U. diversum* GUDIV-103 lipoprotein was expressed in *E. coli* and purified. This is the first report of the production of recombinant proteins from *U. diversum*. rGUDIV-103 stimulated a humoral immunogenic response with the production of specific IgGs against rGUDIV-103 and LAMP extracts from *U. diversum* strains isolated from different regions of Brazil. rGUDIV-103 was also recognised by antibodies against the *U. diversum* strain ATCC 49783. These findings confirmed the immunogenicity and antigenicity of the recombinant protein. Positive Western blotting results for *GUDIV**-103* in PCR-negative strains strongly suggested the occurrence of genetic variation in the sequence of this gene. Recombinant GUDIV-103 increased the production of NO and H_2_O_2_ and expression of caspase-3, the apoptosis marker gene. Low GUDIV-103 doses induced mitogenesis in PBMC cultures. Finally, rGUDIV-103 induced the expression of pro-inflammatory genes (*IL-1β*, *TNF-α*, *TLR2*, and *TLR4*). Thus, this recombinant protein is a promising antigen that may hold potential to be employed in the development of diagnostic tests and prophylactic measures against *U. diversum*.

## Figures and Tables

**Figure 1 microorganisms-10-01032-f001:**
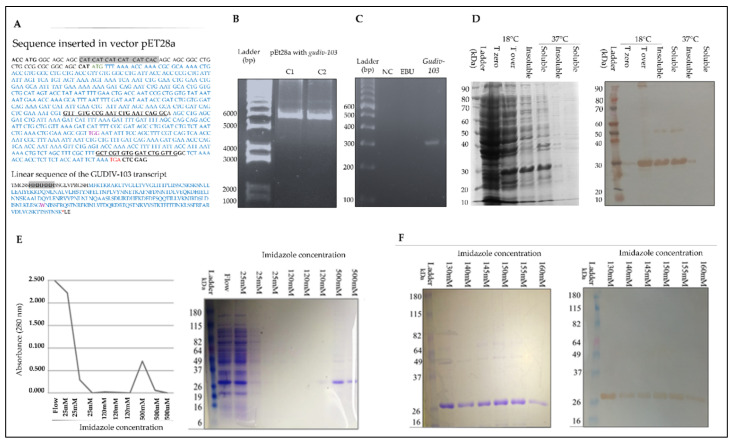
Cloning, heterologous expression, and purification of recombinant GUDIV-103 (rGUDIV-103). (**A**) Top, nucleotide sequence inserted into the pET28a(+) vector; bottom, translation of the inserted sequence. Green, red, grey, and purple nucleotides correspond to the GUDIV translation start codon, stop codon, N-terminal histidine tail, and the TGG codon encoding tryptophan, which replaces the TGA codon in *Escherichia*
*coli*, respectively. The underlined sequences were used to build the forward and reverse primers for assessing gene insertion. (**B**) Agarose gel electrophoresis of plasmids isolated from two different One Shot TOP10 transformants (C1 and C2). (**C**) Agarose gel electrophoresis of PCR-amplified *GUDIV-103* fragment using the primers stated in panel A and DNA extracted from *E. coli* BL21 (DE3) Star transformed with pET-28a(+)*gudiv-103* vector. EBU: untransformed *E. coli* BL21 (DE3) Star; NC: negative control—all elements of the reaction except the target DNA. Ladder: Marker TrackIt 100 bp DNA ladder (Invitrogen, São Paulo, Brazil). (**D**) Assessing different expression temperatures on GUDIV_103 solubility by sodium dodecyl sulphate–polyacrylamide gel electrophoresis (SDS–PAGE) (left) and Western blotting (right). T-zero: total uninduced *E. coli* BL21 (DE3) Star extract; T over: rGUDIV-103 expression in *E. coli* BL21 (DE3) Star after overnight isopropyl β-d-1-thiogalactopyranoside (IPTG) induction. Bands at approximately 27 kDa correspond to GUDIV_103 expression. Insoluble: insoluble fraction of the total bacterial extract after induction; soluble: soluble fraction of the bacterial extract after induction. Ladder: Novex Sharp pre-Stained Protein Standard (Thermo Fisher Scientific). (**E**) Left, monitoring HisTrap purification of GUDIV_103 by spectrophotometry (280 nm) using 25 and 120 mM imidazole wash steps, followed by elution with 500 mM imidazole; right, SDS–PAGE. Flow: eluate before imidazole wash steps. (**F**) Purification and optimisation using 130–160 mM imidazole in the wash step, followed by elution with 500 mM imidazole; analysed by SDS–PAGE (left) and Western blotting (right). Ladder: Novex Sharp protein standard (Invitrogen).

**Figure 2 microorganisms-10-01032-f002:**
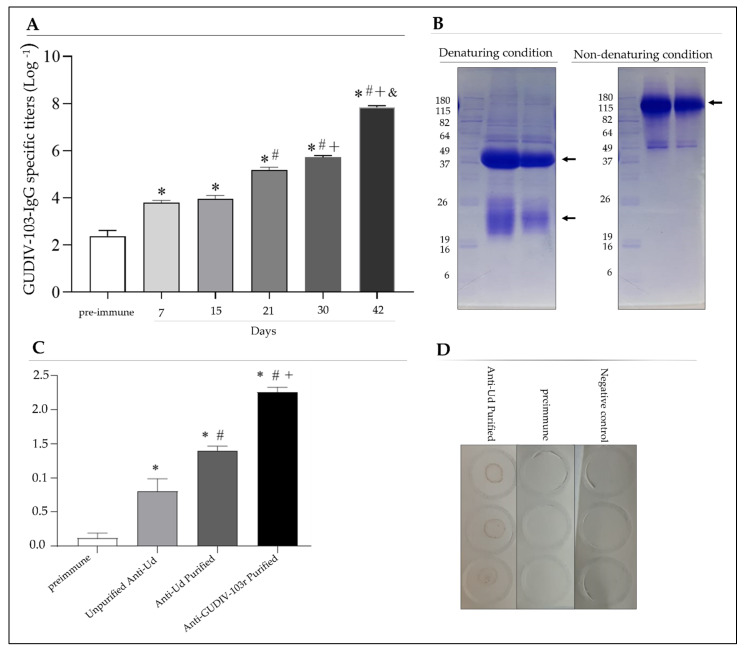
Immunoreactivity of anti-GUDIV-103 antibodies produced by inoculating rabbits with *Ureaplasma diversum* or purified recombinant GUDIV-103 (rGUDIV-103). (**A**) Specific IgG responses induced in rabbits immunised with 500 μg rGUDIV-103 after 7 d (first dose), 15 d (second dose), 21 d (second dose), 30 d (third dose) and 42 d (third dose) or against *U. diversum* ATCC 49783 using blood samples collected every 7 d for 42 d. * *p* < 0.05 vs. preimmune group. *^#^
*p* < 0.05 vs. preimmune group and 7 and 15 days groups. *^#+^
*p* < 0.05 vs. preimmune group and 7, 15 and 21 days groups. *^#+&^
*p* < 0.05 vs. preimmune group and 7, 15, 21 and 30 days groups. (**B**) Electrophoretic profile (12.5% polyacrylamide gel) of specific IgG produced in rabbits after delipidation and purification by affinity chromatography in denaturing (left) and non-denaturing (right) conditions. Both gels were stained using Coomassie blue. Arrows on the left image represent the IgG heavy chain (~50 kDa) and the light chain (~25 kDa), and the arrow in the right image indicates the undenatured IgG antibody (~150 kDa). (**C**) Immunoreactivity of serum samples measured via enzyme-linked immunoassay (ELISA) at 450 nm. The following sera were used: pre-immune, impurified anti-*Ud*, purified anti-*Ud*, and purified anti-rGUDIV-103. rGUDIV-103 (200 ng/well) was used in each well. * *p* < 0.05 vs. preimmune group. *^#^
*p* < 0.05 vs. preimmune group and unpurified anti-Ud group. *^#+^
*p* < 0.05 vs. preimmune group and unpurified anti-Ud group and anti-Ud purified group. (**D**) Dot blot of nitrocellulose membrane sensitised with rGUDIV-103 and incubated with anti-*Ud* serum (1:200), pre-immune serum (1:200), or blocking buffer. Negative control: membrane sensitised with PBS at pH 7.4. Data are expressed as mean ± standard deviation using a one-way analysis of variance with Bonferroni test.

**Figure 3 microorganisms-10-01032-f003:**
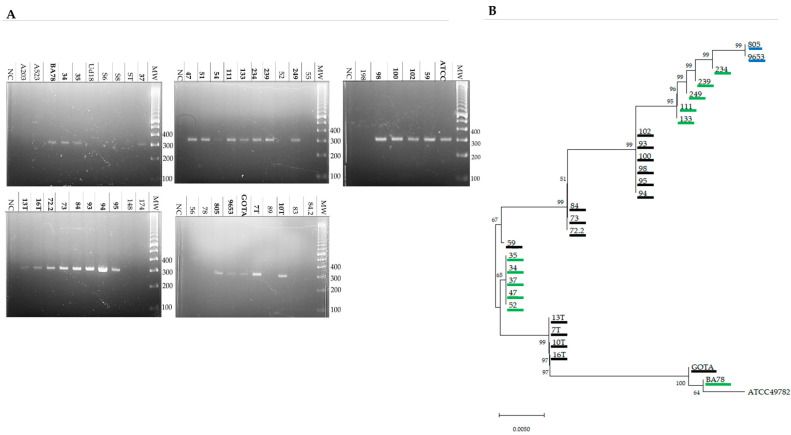
Identification of *GUDIV**-103* from *Ureaplasma*
*diversum* isolates and phylogenetic analysis. (**A**) Agarose gel electrophoresis of PCR-amplified *GUDIV**-103* fragment using DNA extracted from 46 *U. diversum* strains from different Brazilian states. MW: molecular weight marker TrackIt 100 bp ladder (Invitrogen, São Paulo, Brazil). NC (negative control): All elements of the reaction except the target DNA. (**B**) Phylogenetic tree generated using the neighbour-joining method with a Tajima–Nei distance matrix (MEGA-X version 4.1) following the alignment of *GUDIV**-103* sequences from *U. diversum* isolates and the reference strain, ATCC 49782. Strains from Mato Grosso do Sul, Bahia, and São Paulo are underlined in blue, green, and black, respectively.

**Figure 4 microorganisms-10-01032-f004:**
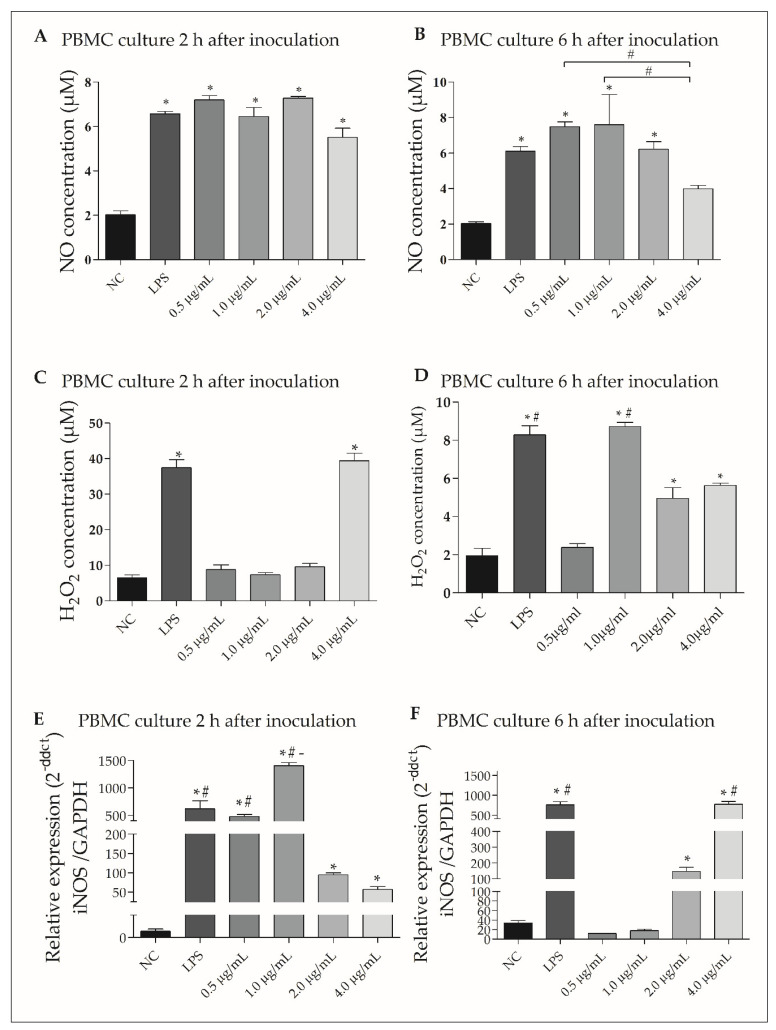
NO and H_2_O_2_ concentrations and inducible nitric oxide synthase (iNOS) expression in bovine peripheral blood mononuclear cell (PBMC) culture supernatant after incubation with 0.5, 1.0, 2.0, and 4.0 µg/mL recombinant GUDIV-103 (rGUDIV-103) lipoprotein for (**A**,**C**,**E**) 2 and (**B**,**D**,**F**) 6 h. (**A**,**B**) NO concentration. (**C**,**D**) H_2_O_2_ concentration. (**E**,**F**) iNOS gene expression. NC: negative control (50 µL phosphate-buffered saline, pH 7.4), positive control: LPS (lipopolysaccharide) (100 ng/mL). Relative iNOS expression compared with that of glyceraldehyde-3-phosphate dehydrogenase (GAPDH). Treatments were compared using the Kruskal–Wallis non-parametric test followed by Dunn’s post hoc test. Different symbols indicate statistically different groups. Statistical significance (*p* < 0.05) is represented by the symbols (∗, #, -). Data are expressed as the mean ± standard deviation (n = 9). * *p* < 0.05 vs. negative group. ^#^
*p* < 0.05 vs. LPS group. ^-^
*p* < 0.05 vs. 0.5 μg/mL group.

**Figure 5 microorganisms-10-01032-f005:**
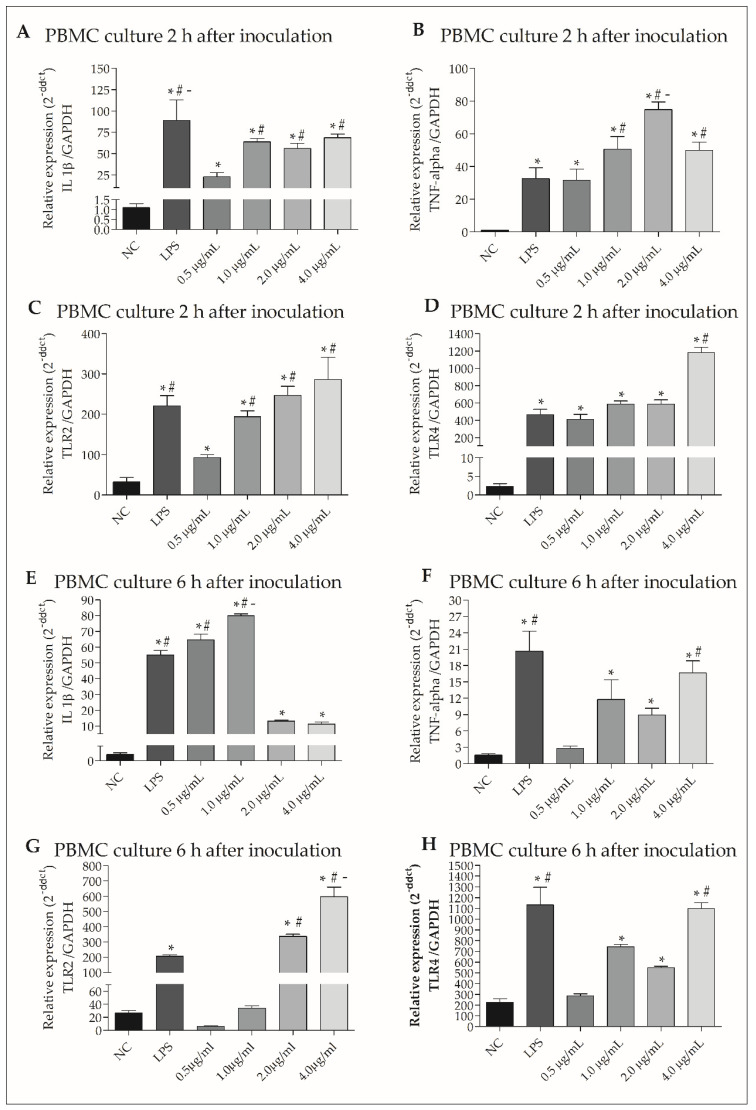
Gene expression of cytokines (interleukin 1 beta (IL-1β), tumour necrosis factor alpha (TNF-α), and toll-like receptor (TLR)) in bovine peripheral blood mononuclear cells (PBMCs) incubated for 2 h with 0.5, 1.0, 2.0, and 4.0 µg/mL recombinant GUDIV-103 (rGUDIV-103) lipoprotein for (**A**–**D**) 2 and (**E**–**H**) 6 h. (**A**,**E**): *IL-1β*. (**B**,**F**): *TNF-α*. (**C**,**G**): *TLR2*. (**D**,**H**): *TLR4*. The negative control (NC) and positive control were PBS (pH 7.4) and LPS (100 ng/mL), respectively. Treatments were compared using the Kruskal–Wallis non-parametric test followed by Dunn’s post hoc test. Different symbols indicate statistically different groups. Statistical significance (*p* < 0.05) is represented by the symbols (∗, #, -). Data are expressed as the mean ± standard deviation (n = 9). * *p* < 0.05 vs. negative group. ^#^
*p* < 0.05 vs LPS group. ^-^ *p* < 0.05 vs. 0.5 μg/mL group.

**Figure 6 microorganisms-10-01032-f006:**
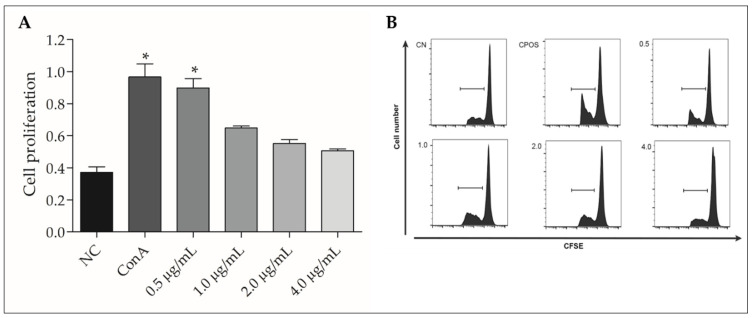
Lymphoproliferation assay of bovine peripheral blood mononuclear cells (PBMCs) stimulated with 0.5, 1.0, 2.0, and 4.0 µg/mL recombinant GUDIV-103 (rGUDIV-103) antigen. (**A**) rGUDIV-103 stimulated cell division index. X-axis includes the different antigen concentrations, the negative control (NC; PBS 1X, pH 7.4), and positive control (concanavalin A, ConA). Statistical significance (*p* < 0.05) is represented using *. (**B**) Detection of lymphocyte division using fluorescence intensity. The arrow above 5(6)-carboxyfluorescein diacetate succinimidyl ester (CFSE) represents its emission spectrum, with the horizontal line indicating the region (peak area) analysed. Treatments were compared using the Kruskal–Wallis non-parametric test, followed by Dunn’s post hoc test. Data are expressed as the mean ± standard deviation (n = 9). * *p* < 0.05 vs negative group.

**Figure 7 microorganisms-10-01032-f007:**
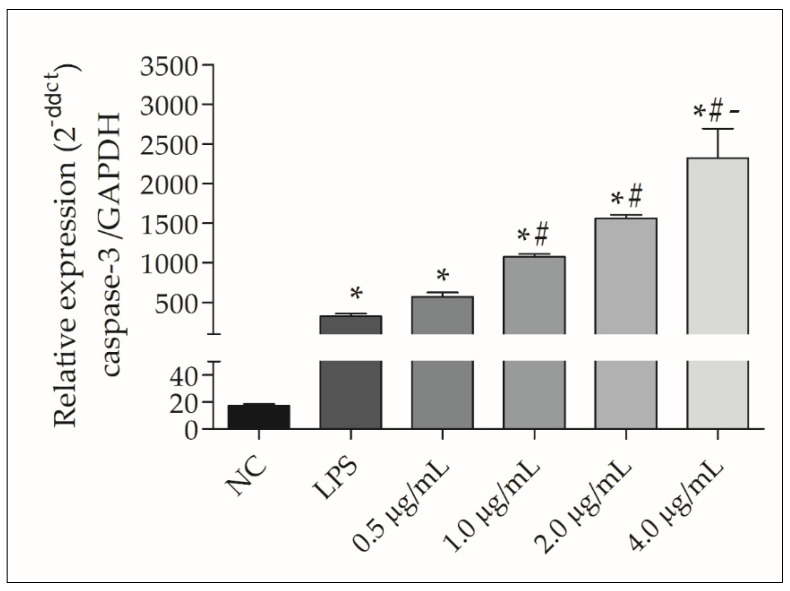
Caspase-3 expression in bovine peripheral blood mononuclear cells (PBMCs) treated with 0.5, 1.0, 2.0, and 4.0 µg/mL recombinant GUDIV-103 (rGUDIV-103) for 6 h. The negative control (NC) and positive control were PBS at pH 7.4 and LPS (100 ng/mL), respectively. Treatments were compared using the Kruskal–Wallis non-parametric test followed by Dunn’s post hoc test. Different symbols indicate statistically different groups. Statistical significance (*p* < 0.05) is represented by the symbols (∗, #, -). Data are expressed as the mean ± standard deviation (n = 9). * *p* < 0.05 vs negative group. ^#^
*p* < 0.05 vs. LPS group. ^-^
*p* < 0.05 vs 0.5 μg/mL group.

**Table 1 microorganisms-10-01032-t001:** PCR and Western blotting of *Ureaplasma*
*diversum* from different Brazilian states.

Brazilian State	*U. diversum* Strain	PCR	Western Blotting	
			Hydrophobic Phase (UdLAMPs)	Hydrophilic Phase (Cytosolic Proteins)
**BA**	BA78	+	+	−
	34	+	+	−
	35	+	+	−
	37	+	+	−
	47	+	+	−
	51	+	+	−
	52	−	+	−
	54	+	+	−
	55	−	+	−
	56	−	+	−
	78	−	+	−
	84.2	−	+	−
	89	−	+	−
	111	+	+	−
	133	+	+	−
	148	−	+	−
	174	−	+	−
	198	−	+	−
	234	+	+	−
	249	+	+	−
**MS**	805	+	+	−
	9653	+	+	−
**SP**	A203	−	+	−
	GOTA	+	+	−
	S6	−	+	−
	S8	−	+	−
	5T	−	+	−
	7T	+	+	−
	10T	+	+	−
	13T	+	+	−
	16T	+	+	−
	72.2	+	+	−
	73	+	+	−
	84	+	+	−
	93	+	+	−
	94	+	+	−
	95	+	+	−
	98	+	+	−
	100	+	+	−
	59	+	+	−
**#**	A523	−	+	−
	ATCC49782	+	+	−

BA: Bahia, MG: Minas Gerais, MS: Mato Grosso do Sul, SP: São Paulo. Strain # A523 had no defined locations.

## Data Availability

The datasets used and/or analyzed during the current study are available from the corresponding author on reasonable request.

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
