# Peer review of "Heterologous Expression, Purification, and Immunomodulatory Effects of Recombinant Lipoprotein GUDIV-103 Isolated from Ureaplasma diversum"

_microorganisms, 2022, doi:10.3390/microorganisms10051032_

Round 1

Reviewer 1 Report

The manuscript presents interesting data about production and immunomodulatory effects of the lipoprotein isolated from the important pathogen from the Mollicutes class responsible for genital and reproductive disorders in cattle. In general, the manuscript is well written, the study was well designed and the results were generally well interpreted.

Detailed comments

Introduction

Lines 42-44: the authors should indicate if U. diversum requires the presence of CO2 and other gases to growth, if so, please specify their concentration

Line 72: the authors should add ‘s’ to ‘Mollicute’

Lines 77-80: the authors should provide more data on the lipoprotein, if needed please complete the references

Materials and Methods

Line 96: the authors should indicate the most important compositions of the UB medium, is it commercial or in-house?; if commercial – please add specification of the product

Line 212: the reviewer have doubts about too few animals (two) used in the study, please justify it

Lines 234-246: Is it in-house ELISA? Was it validated before the study? Please add details about

Line 254: the authors should more specify which samples were used here

General comment: why were PBS-T and PBS used interchangeably in the presented methods?

Line 347: was is the same temperature as above mentioned?

Results

Figure 1E is not readable enough

Lines 412-415: two first sentences should be moved to the Materials and Methods section

Figure 3 – there are no marked colours as was indicated in the figure legend

Line 553 – there should be information about pH of PBS instead ‘_’

Author Response

Response to Reviewer 1 Comments

We are thankful for all reviewer comments and suggestions. The peer-review pointed out important aspects of our manuscript and contributed to improving its quality. We are in agreement with the observations that have been done for this editorial group and modifications in the submitted manuscript are presented below

Point 1: Lines 42-44: the authors should indicate if U. diversum requires the presence of CO and other gases to growth, if so, please specify their concentration.

Response 1: Thank you very much for your remark. We have added the CO2 concentration (8 to 10%) in the new version of the manuscript.

Point 2: Line 72: the authors should add ‘s’ to ‘Mollicute’

Response 2: Thank you very much for your remark. We corrected it in the new version of the submitted manuscript.

Point 3: Lines 77-80: the authors should provide more data on the lipoprotein, if needed please complete the references.

Response 3: Thank you very much for your remark. We have provided more details about lipoprotein (lines 78-86) in the new version of the manuscript.

Point 4: Line 96: the authors should indicate the most important compositions of the UB medium, is it commercial or in-house?; if commercial – please add specification of the product.

Response 4: Thank you very much for your remark. We have provided more details about UB medium in lines 102-104 in the new version of the manuscript.

Point 5: Line 212: the reviewer have doubts about too few animals (two) used in the study, please justify it

Response 5: Thank you very much for your remark. In this study, our main objective was to evaluate the immunogenicity of recombinant anti-gudiv-103 antibodies. We did not aim to compare responses between different animals. Therefore, we planned so that at the end of the antibody production process there would be enough serum to carry out the experiments, in this we concluded that two animals would provide the amount of serum needed for this study.

Point 6: Lines 234-246: Is it in-house ELISA? Was it validated before the study? Please add details about.

Response 6: We added more information about performing the ELISA and added the citation that served as the basis for this experiment. New information has been added on lines 259-260 of the new version.

Point 7: Line 254: the authors should more specify which samples were used here.

Response 7: Thank you very much for your remark. In summary rGUDIV-103 protein was applied to a 0.45 µm nitrocellulose membrane and incubated with pre-immune and hyper-immune serum produced against rGUDIV-103 or serum pro-duced against U. diversum ATCC 49783. This information was added in lines 267 -271 of the new version of the manuscript.

Point 8: General comment: why were PBS-T and PBS used interchangeably in the presented methods?

Response 8: We used PBS-T in methodologies that involved blocking or washing nitrocellulose membranes or ELISA plates (in immunoassays) to prevent binding of specific antibodies to the plate or membrane sensitized with rgudiv-103. In the methodologies that did not involve blocking or lavgagen, only PBS was used.

Point 9: Figure 1E is not readable enough

Response 9: Thank you very much for your remark. We have improved the quality of the figure 1E and submitted it in the new version of the manuscript.

Point 10: Lines 412-415: two first sentences should be moved to the Materials and Methods section

 Response 10: Thank you very much for your remark. We removed these sentences from the referring section (line 417 in the new version of the manuscript )as they were already stated in the Materials and Methods section.

Point 11: Figure 3 – there are no marked colours as was indicated in the figure legend.

Response 11: Thank you very much for your remark. We have improved the quality of figure 3 and submitted it in the new version of the manuscript.

Point 12: Line 553 – there should be information about pH of PBS instead

Response 12: Thank you very much for your remark. The correct pH value (7.4) has been added (L-554 of the new version)

We reinforce our acknowledgments for all comments and we hope that this revised manuscript could be published by this editorial group.

Yours sincerely,

Lucas Marques

Reviewer 2 Report

The manuscript submitted by Santos-Junior et al. entitled “Heterologous expression, purification, and immunomodulatory 2 effects of recombinant lipoprotein gudiv_103 isolated from Ureaplasma diversum” presents an account of successful recombinant protein expression in E. coli and use of the extracted protein for several immune-mediated responses.

In Line 282 of the methods it states “10 cells per mL” this seems very low as a target – is there a superscript number missing here?

Figure 1 and 3 are not appropriately inserted into the template. I had to copy and paste them into powerpoint to see the right half of the figures at all.

The avidity assay is unnecessary in Figure 2 as there is no comparison to any controls or standards. GIT will denature all proteins and block immunoreactivity eventually. The accepted method of determining antibody avidity (polyclonal) or affinity (monoclonal) is through the method of surface plasmon resonance. Equally as the polyclonal antibodies would be partially directed against the histag – a more relevant investigation would be to determine the component of the polyclonal immune response that is directed against the histag component, or residual reactivity following pre-adsorption against histag or an irrelevant histag containing protein.

The phylogenetic analysis presented in Figure 2B appears to show separation of two distinct clades, 805 and 9653, are very distant from the ATCC strain and yet the PCR primers still amplify these distant relatives.  More analysis of these far distant gene variants is warranted, as the phylogenetic analysis does not give a sense of protein length or number of non-synonymous polymorphisms between these outlying gene variants.

The data contained in Table 1 is of concern. There are no negative western blot results, irrespective of whether the strain contained a PCR positive or not.  Equally the authors indicated in the phylogenetic analysis that there were insertions and deletions in the sequences of the gene for some of the strains. Did this result in truncation of the gene?  Specificity of the Western blot results would be increased if corresponding changes in protein size were observed for strains shown to have corresponding insertions and deletions.  Given that the polyclonal antibody also will have reactivity to the histag – proteins with highly charged motifs may also react with the antibody. Perhaps control blots incubated with a commercial polyclonal anti-histag antibody would also serve as a control for wild type strain reactivity.

The immune response data is very compelling and interesting. However, the proteins are lipoproteins and will form micelles on purification. Have the authors performed LPS assays on their extracts (there are several good commercial kits available), to ensure that LPS is not being carried along by the purification process that would give a background activation.  The authors cite previous reports looking at NO and H2O2 production with LAMP extracts or recombinant LAMPs (line 628) -however neither of these references use recombinant proteins, in fact they both use incubation with whole centrifuged organisms. One of the limitations of this study is the lack of a comparative control: it is unclear if the effects are being mediated by protein in combination with the lipid anchor, or if any lipid-anchored protein would mediate this effect. The latter is further complicated by the fact that the attached lipids in this particular case are derived from E.coli and not from Ureaplasma diversum, which poses the question of whether lipids derived from U. diversum would have the same effects.  While the assays performed by the authors appear to be systematic in their investigation, a comparison to the amplitude of the effects relative to whole U. diversum or perhaps Tx114 extracts similar to those used in citation 78, would have built significant confidence in the Mollicute-related component of the hypothesis. These limitations to the conclusions that can be drawn need to be discussed thoroughly in the discussion. A mutant lacking the lipid-anchor signal run in parallel to these studies, would have been my preferred control.

Furthermore, if GUDIV-103 is only one of the many LAMPs produced by U. diversum, it would also be important to determine if immunological pressure would result in the loss of expression of this protein (with minimal fitness cost to the bacterium) as has been found for many lipid-anchored proteins in other Mollicutes, before this protein can be proposed as a really as strong a candidate for vaccination-based immunotherapy or prophylactic approach for herd protection.

Author Response

Response to Reviewer 1 Comments

We are thankful for all reviewer comments and suggestions. The peer-review pointed out important aspects of our manuscript and contributed to improving its quality. We are in agreement with the observations that have been done for this editorial group and modifications in the submitted manuscript are presented below

Point 1: In Line 282 of the methods it states “10 cells per mL” this seems very low as a target – is there a superscript number missing here?.

Response 1: Thank you very much for your remark. We have added the correct number (105) in the new submitted manuscript version.

Point 2: Figure 1 and 3 are not appropriately inserted into the template. I had to copy and paste them into powerpoint to see the right half of the figures at all.

Response 2: Thank you very much for your remark. We have adjusted the position of figures in the new submitted version.

Point 3: The avidity assay is unnecessary in Figure 2 as there is no comparison to any controls or standards. GIT will denature all proteins and block immunoreactivity eventually. The accepted

method of determining antibody avidity (polyclonal) or affinity (monoclonal) is through the method of surface plasmon resonance. Equally as the polyclonal antibodies would be partially directed against the histag – a more relevant investigation would be to determine the component of the

polyclonal immune response that is directed against the histag component, or residual reactivity following pre-adsorption against histag or an irrelevant histag containing protein.

Response 3: We agree with your observations, your comments in line with the literature on the subject, made it clear that the proposed avidity test is not suitable in this work, therefore we removed it from the analysis.

Point 4: The phylogenetic analysis presented in Figure 2B appears to show separation of two distinct clades, 805 and 9653, are very distant from the ATCC strain and yet the PCR primers still amplify these distant relatives. More analysis of these far distant gene variants is warranted, as the phylogenetic analysis does not give a sense of protein length or number of non-synonymous polymorphisms between these outlying gene variants.

Response 4: We do not believe that application of gudiv-103 from strains from such distant clades is related to gene conservation for gudiv-103 in U. diversum. We are conducting further research to clarify these findings, but at this time our data do not allow us to carry out more in-depth analyzes on the topic. So we make this limitation clear in the new version of the manuscript (line 607-609).

Point 5: The data contained in Table 1 is of concern. There are no negative western blot results, irrespective of whether the strain contained a PCR positive or not. Equally the authors indicated in the phylogenetic analysis that there were insertions and deletions in the sequences of the gene for some of the strains. Did this result in truncation of the gene? Specificity of the Western blot results would be increased if corresponding changes in protein size were observed for strains shown to have corresponding insertions and deletions. Given that the polyclonal antibody also will have reactivity to the histag – proteins with highly charged motifs may also react with the antibody. Perhaps control blots incubated with a commercial polyclonal anti-histag antibody would also serve as a control for wild type strain reactivity..

Response 5: Thank you very much for your remark. Table 1 refers to total proteins extracted from different strains of U. diversum. In table 1 the positive data were for the hydrophobic portion of the extract of total lipoproteins from different strains of U. diversum, as the proteins used are extracted directly from the strains. do not present the histin syrup. We believe that positive Western blotting results even for PCR negative strains may be related to gene alterations that do not lead to substantial changes in the primary structure of lipoproteins. Thus, the protein epitopes would be preserved even with some alterations at the gene level. We have made this information clearer in the new manuscript version (lines 595-599).

Point 6: The immune response data is very compelling and interesting. However, the proteins are lipoproteins and will form micelles on purification. Have the authors performed LPS assays on their extracts (there are several good commercial kits available), tomensure that LPS is not being carried along by the purification process that would give a background activation. The authors cite previous reports looking at NO and H O production with LAMP extracts or recombinant LAMPs (line 628) -however neither of these references use recombinant proteins, in fact they both use incubation with whole centrifuged organisms. One of the limitations of this study is the lack of a comparative control: it is unclear if the effects are being mediated by protein in combination with the lipid anchor, or if any lipid-anchored protein would mediate this effect The latter is further complicated by the fact that the attached lipids in this particular case are derived from E.coli and not from Ureaplasma diversum, which poses the question of whether lipids derived from U. diversum would have the same effects. While the assays performed by the authors appear to be systematic in their investigation, a comparison to the amplitude of the effects relative to whole U. diversum or perhaps Tx114 extracts similar to those used in citation 78, would have built significant confidence in the Mollicute-related component of the hypothesis. These limitations to the conclusions that can be drawn need to be discussed thoroughly in the discussion. A mutant lacking the lipid-anchor signal run in parallel to these studies, would have been my preferred control.

Response 7: Thank you very much for your remark. In the present study we also performed the analysis of NO production from cells treated with total extract of lipoproteins extracted from U. divesum , and by the viable or inactivated strain. In all cases there was an increase in the production of NO (we have inserted these data as supplementary material in the new version of the manuscript - L351-353). We believe that these results, together with the NO H2O2 assay, show that GUDIV-103 may play an important role in the induction of these molecules during U. diversum infection.

We fully agree with the reviewer regarding the lack of a mutant lacking the lipid-anchor signal run in parallel to these studies, could make the results on immunomodulation more robust. Therefore, we make this limitation clear in the new version of the manuscript (lines 644-647).

Point 7: Furthermore, if GUDIV-103 is only one of the many LAMPs produced by U. diversum, it would also be important to determine if immunological pressure would result in the loss of expression of this protein (with minimal fitness cost to the bacterium) as has been found for many lipid-anchored proteins in other Mollicutes, before this protein can be proposed as a really as strong a candidate for vaccination-based immunotherapy or prophylactic approach for herd protection.

Response 2: Thank you very much for your remark. We do not make it clear in the new version that studies evaluating whether immunological pressure would result in loss of expression of this protein (with minimal cost of adaptation to the bacteria) should be carried out to better clarify the potential immunobiological application of GUDIV-103 (line 688-890).

We reinforce our acknowledgments for all comments and we hope that this revised manuscript could be published by this editorial group.

Yours sincerely,

Lucas Marques